# Singular sublimation of ice and snow crystals

Etienne Jambon-Puillet [1], Noushine Shahidzadeh [1] & Daniel Bonn [1]

The evaporation (sublimation) of ice and snow has a major impact on global climate, since the amount of ice and snow determines Earth's albedo. Yet, due to their complex geometry with several sharp regions which are singular for the evaporation, the precise evaporation dynamics of snow and ice crystals remains challenging to predict. Here, we study the sublimation of snowflakes and pointy ice drops. We show that the evaporation rates of water and ice drops are similar; they are both limited by the diffusive transport of the vapour. This allows us to predict ice and snowflake evaporation quantitatively by solving the diffusive free-boundary problem, which correctly predicts the rapid self-similar evolution of sharp edges and points. Beyond providing a conceptual picture to understand the sublimation of ice crystals, our results are more generally applicable to other diffusion problems such as the dissolution of salt crystals or pharmaceuticals.

[1] Institute of Physics, Van der Waals-Zeeman Institute, University of Amsterdam, Science Park 904, 1098 XH Amsterdam, The Netherlands. Correspondence and requests for materials should be addressed to E.J.-P. (email: e.a.m.jambonpuillet@uva.nl)

Solids generally do not evaporate in ambient air. Common sense thus suggests that ice also does not rapidly evaporate (or sublimate) on Earth. Yet, ice and snow do in fact evaporate significantly in ambient air; a thin layer of snow can be seen to disappear in typically a few days, even if the temperature stays below freezing. Understanding the formation and subsequent evolution of ice is crucial in various problems: de-icing[1], freeze-drying[2], glaciology[3,4] and to better understand the climate of planets, such as the Earth or Mars[5–8]. As such, intense ongoing research efforts are made to unravel water freezing[9–14]. Yet, the evaporation (sublimation) of ice crystals, although very common, has remained largely unexplored. Previous studies on evaporating polygonal ice crystals have shown that the crystal edges, which are sharper, were the first to recede. This has been previously attributed to defects in the crystaline structure at the crystal edge[15]; however no quantitative measurements or models exist of the full evaporation process.

Here we demonstrate that contrary to crystal growth and unlike what was previously thought[15], the shape evolution of sublimating ice and snow crystals has nothing to do with the underlying crystal structure. We show that the rounding of sharp edges and points is the direct consequence of the singular nature of the evaporation mechanism through diffusion into the vapour phase close to regions of high curvature and describe its dynamics quantitatively for the first time. For liquid drops, the local diffusion-limited evaporation rate diverges at the contact line (a region of infinite curvature). If the drop contact line stays pinned, an outward flow in the drop then replenishes the corner region, driving the well-known "coffee-stain" effect[16–18]. As snowflakes have many sharp points, we anticipate the evaporation to be locally stronger there. However, because the mass loss due to evaporation cannot be replenished in such a solid body, evaporation at the sharp points makes them recede first, resulting in their self-similar smoothing.

## Results

**Snowflake sublimation.** Subsequent experimental images of a single snowflake evaporating in dry air are shown in Fig. 1a[19]. Like polygonal ice crystals[15], the outward pointing protrusions of the crystal structure disappear first, suggesting that the evaporation is enhanced at sharp regions and proceeds from the outside towards the inside. To understand this dynamic evaporation

pattern, we first focus on the somewhat simpler case of pointy ice droplets that exhibit an isolated singular point and subsequently consider the snowflakes (Fig. 1b).

**Pointy ice drop experiment.** In a chamber of constant low humidity (RH ≈ 5%), we deposit small water drops on a cold substrate (at constant temperature $T_s$), freeze them and monitor their evaporation (see Methods, Fig. 2 and Supplementary Movie 2). In a typical experiment, the liquid drop reaches $T_s < 0$ °C in a few seconds, but can remain supercooled for several minutes while evaporating at the same time. Subsequently, the drop solidifies and due to the expansion of water upon freezing, a sharp conical tip forms at its top[20,21]. This pointy ice drop then continues to evaporate (see Supplementary Movie 3). Figure 3 shows the volume of such a drop as a function of time. Perhaps contrary to intuition, the ice drop volume decreases at almost the same rate as the liquid drop; at first glance, water and solid ice evaporate at the same rate. However, significant differences are observed for the evolution of the drop shape (see Fig. 2 and Supplementary Movie 3). The liquid drop evaporates with a strongly pinned contact line in the usual constant contact radius mode[22], while the frozen drop radius clearly decreases during evaporation. The drop's contact line retracts and consequently, the evaporation slows down over time[22,23].

Three observations can be made from these experiments. First, the fact that the supercooled liquid contact line stays pinned, while the ice contact line retracts is surprising, since the anchoring of the ice drop to the rough surface is much stronger; one needs a large force to detach the frozen drop from the plate. Second, the evaporation of ice and supercooled water occurring at roughly the same speed suggests that the evaporation of ice is limited by the diffusion of water molecules in the vapour phase, as is known to be the case for liquid drops. This can be explained by the fact that ice and supercooled water molecules have similar properties[24] and a similar volatility[25]. Third, and most importantly, the evaporation of the pointy drop is qualitatively similar to that of the snowflakes: regions of high curvature, in this case the tip and contact line, are the first to disappear and the surface smoothens in time (Fig. 2, Supplementary Movies 2 and 3).

**Modelling framework.** The early stage of evaporation is therefore dominated by the regions of high curvature. It is important to

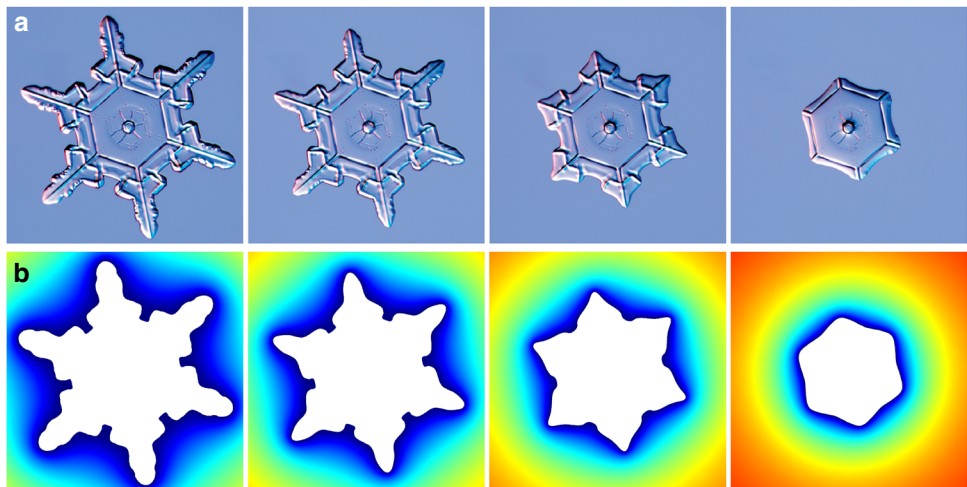

**Fig. 1** Snowflake sublimation. **a** Sequence of images showing an evaporating snowflake (adapted from[20], Copyright © 2007, 2010 by Kenneth Libbrecht). **b** Snapshots of a 3D simulation of the same snowflake assuming uniform thickness with typical parameters: radius 1.1 mm, thickness 0.2 mm, $T_s = -10$ °C, RH = 0%. Colours denote the vapour mass concentration $\rho$ ranging from the saturation concentration $\rho = \rho_{sat}$ (blue) to $\rho = 0$ (red); see Supplementary Movie 1

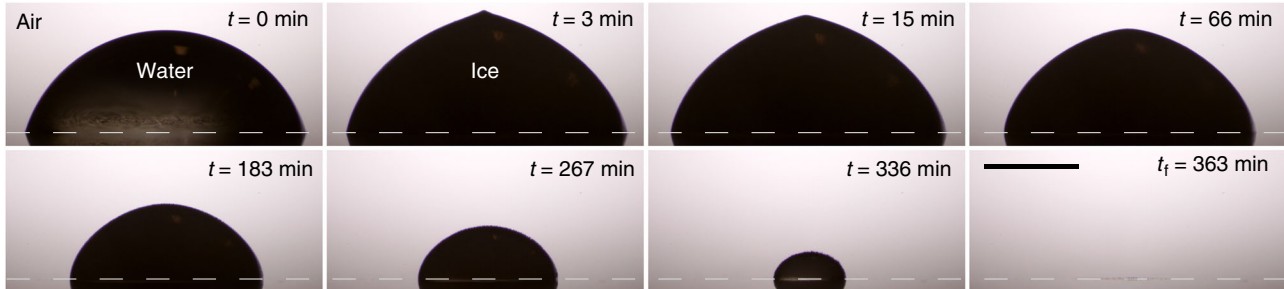

**Fig. 2** Evaporation of a pointy ice drop. Sequence of images documenting the evaporation in air of a pointy ice drop of initial volume $V_0 = 4.4\,\mu L$ on a cold surface ($T_s = -10\,°C$, RH = 4.8%; see Supplementary Movie 2). The pictures include the drop's reflection on the substrate surface (highlighted by the dashed line). Scale bar 1 mm

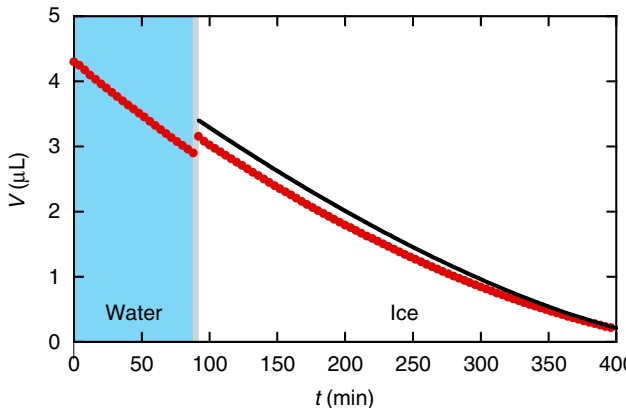

**Fig. 3** Water vs. ice evaporation. Volume as a function of time for a water drop evaporating on a cold plate ($T_s = -15\,°C$, RH = 2.8%). The drop is initially supercooled (blue background) and freezes after 92 min (white background, see Supplementary Movie 3). Red symbols are experimental results, the black curve is the result of the analytical model (see Supplementary Note 2)

understand this stage, since the snowflake retains regions with large curvatures almost through the whole evaporation process. We therefore model ice sublimation by solving the diffusion equation around the complex crystal geometries and because there is no replenishing flows in solids, the receding velocity at every point of the interface is completely determined by the local evaporative flux. Since the ice and water evaporation rates are similar, we assume as for liquid drops[16–18,23] that ice evaporation is slow enough to be quasi-static, and that the vapour mass concentration $\rho$ diffuses with a constant diffusion coefficient $D$ from its (constant) saturation value $\rho_{sat}$ at the ice surface to a fixed (lower) value $\rho_\infty$ far away from the ice (see Methods). The problem then reduces to solving the coupled system of equations

$$\nabla^2 \rho(\mathbf{r}) = 0, \quad \rho|_{\mathbf{r}=\Omega} = \rho_{sat}(T_s), \quad \rho|_{\mathbf{r}=\infty} = \rho_\infty \quad (1)$$

$$v_n \equiv j_n = -\frac{D}{\rho_{ice}} \mathbf{n} \cdot \nabla \rho(\mathbf{r})|_{\mathbf{r}=\Omega}, \quad (2)$$

where $\rho_{ice}$ denotes the ice density, $\Omega$ the solid boundary whose outward normal is $\mathbf{n}$ and $v_n$ and $j_n$ the speed and local (volume) flux of the interface, respectively, in the normal direction.

Since Eq. (1) is Laplace's equation, there is a strong analogy with electrostatics: our diffusion problem for $\rho(\mathbf{r})$ is mathematically equivalent to the electric potential around a charged conductor[16,17,22]. The strong curvature dependence we observe is thus analogous to the known electrostatic tip effect. For a charged

cone of semi-angle $\alpha < 90°$, the electric field at a distance $r$ from the tip apex scales as $E \sim r^{\nu-1}$ with $\nu \approx \frac{2.405}{\pi-\alpha} - \frac{1}{2}$ and thus diverges at the apex ($r = 0$)[26]. Since the electric field is mathematically equivalent to the local evaporation rate $j$, we thus expect a diverging flux at the tip and thus its smoothing, as seen on our ice drops. Considering a single corner yields similar results[26] and since regions of high curvature locally consist of tips and corners, the above arguments are general. However, contrary to the previous literature on electrostatic tips and pinned liquid drops, here the interface is free to move in all directions and the tip shape does not remain conical. Therefore, these arguments are only qualitative, and Eq. (1) must be fully coupled to Eq. (2) to capture the dynamics quantitatively, making the problem much more difficult; it cannot be solved analytically for complex geometries such as the snowflake.

**Initial evaporation of singular tips**. Figure 4a shows the tip profiles of an ice drop during the first half hour of an experiment, where most of the smoothing occurs. These profiles are self-similar as shown in Fig. 4b; they collapse on an hyperbola once rescaled by the curvature $\kappa$. Because the tip profile is self-similar, the full-tip shape during the smoothing process is solely given by one (time-dependent) parameter: the curvature $\kappa(t)$. The self-similarity of the tip shape then allows us to also derive a similarity solution that gives the scaling of the curvature in time by solving Eqs. (1) and (2) for an hyperboloidal tip (see Supplementary Note 1 and Supplementary Figures 1, 2 and 3). Such singularities are characterised by a universal scaling that is independent of the initial or boundary conditions, and in our case the singularity is uniquely determined by the diffusion equation, giving the following scaling for the curvature:

$$\kappa(t) = A(t + t_0)^{-1/2}, \quad (3)$$

where $t_0 = (A/\kappa(0))^2$ denotes the distance to the singularity at the beginning of the experiment. We extract the tip curvatures at the beginning of a typical experiment and plot it in Fig. 4c and d. The data perfectly follow the $-1/2$ power law of Eq. 3.

Our similarity solution gives the dynamics of tip rounding and should not be confused with a possible self-similar (fractal) structure of the snowflakes, which is only a way to quantify the geometry and says nothing about the dynamics. The smoothing mechanism presented here is also original, as sharpening is usually reported for seemingly similar free-boundary problems governed by different physical mechanisms[3,4,27,28]

**Late time evaporation of smooth drops**. For the late stages of evaporation, after about half of the total evaporation time $t_f$, the ice tip has completely disappeared (Fig. 2). Similarly, the drop edge is also smoothed and the apparent contact angle $\theta$ rapidly

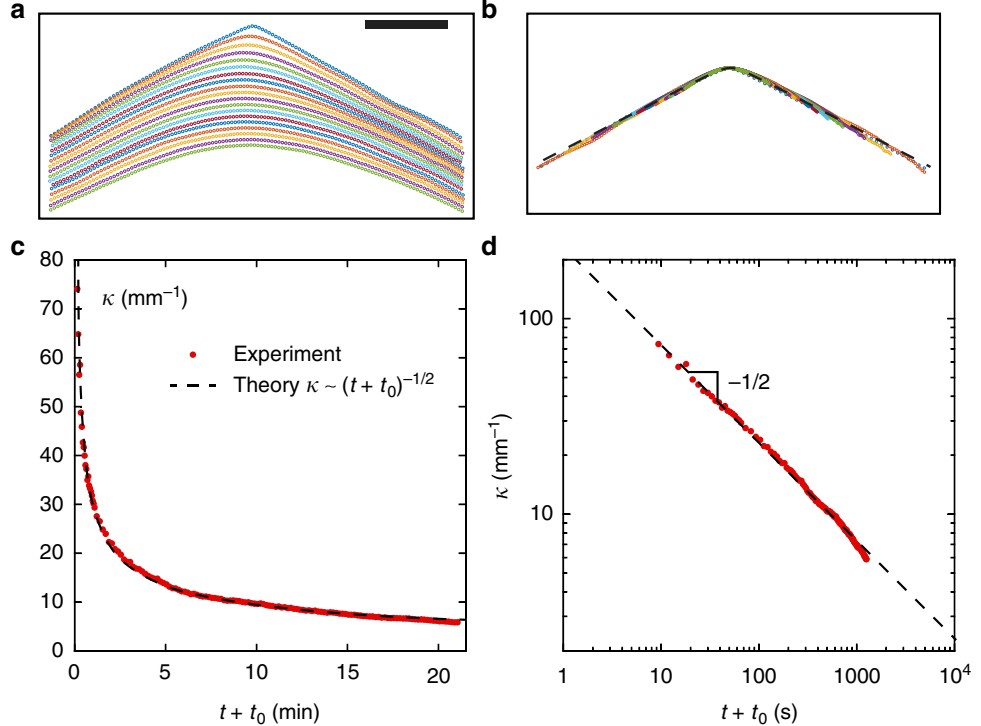

**Fig. 4** Tip smoothing. **a** Experimental tip profiles taken every 2 min during the first half hour of an experiment (see Supplementary Movie 4; $V_0 = 3.5\,\mu L$, $T_s = -12\,°C$, RH = 5.1%). Scale bar 100 μm. **b** Same profiles multiplied by the tip curvature $\kappa$. The dashed black line is a hyperboloid. **c** Tip curvature at short times ($T_s = -15\,°C$, RH = 4.6%). The dashed line is a fit of our similarity solution: $\kappa = A(t + t_0)^{-1/2}$ with $A = 2.32 \cdot 10^5\,s^{1/2}m^{-1}$ ($t_0 = (A/\kappa(0))^2 = 9\,s$). **d** Same data on a log-log scale

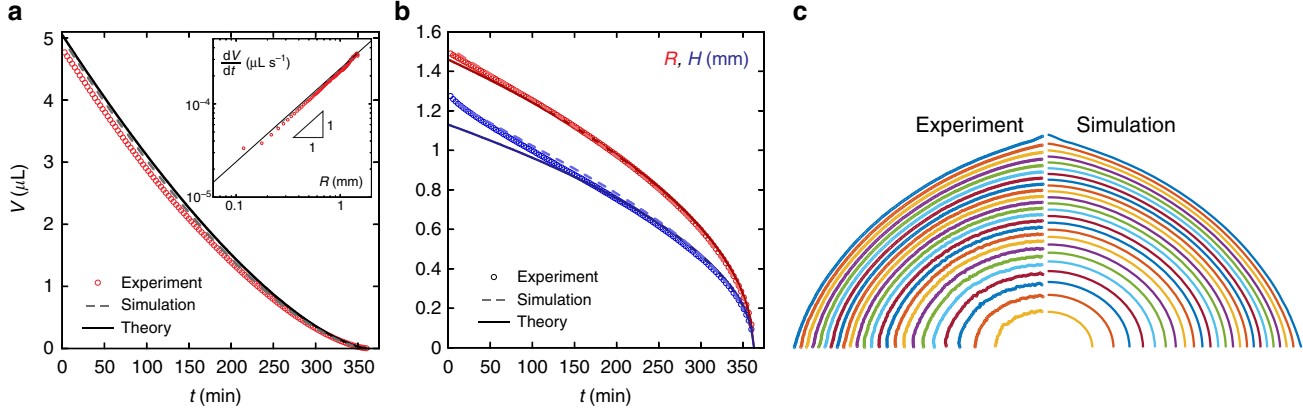

**Fig. 5** Global ice drop evaporation. **a** Evolution of the volume $V$ of the drop shown in Fig. 2; the inset shows the instantaneous volume flux $dV/dt$ as a function of the drop radius. **b** radius $R$ (red circles) and height $H$ (blue circles) for the same drop. The dashed and solid lines denote the numerical simulation and analytical results, respectively (see Supplementary Note 2). **c** Experimental and numerical profiles taken every 15 min for the same drop

evolves towards 90° (see Supplementary Figure 4). A phenomenon very similar to the tip smoothing thus also occurs at the drop edge but in two dimensions (neglecting the much larger drop curvature). We thus expect a similar derivation considering hyperbolas instead of hyperboloids to be able to capture the phenomenon quantitatively (as there should be differences since the geometry is different).

We show in Fig. 5 the global evolution of the volume and shape of the drop presented in Fig. 2. The total evaporation flux, shown in inset of Fig. 5a, scales with the drop radius and not its surface area confirming the purely diffusive picture[29]. Repeating the experiment on substrates with different wettability ($20° < \theta < 120°$) yields similar results: the drop always reaches a smooth self-

similar spheroidal shape whose aspect ratio depends on $\theta$ (see Supplementary Figure 5). Again, because the shape is self-similar, we can analytically solve the diffusion-limited sublimation problem, Eqs. (1) and (2), for general ellipsoids (see Supplementary Note 2 and Supplementary Figure 6). We recover the same scalings as for evaporating liquid drops without a pinned contact line[22], droplet radius $R \sim (t_f - t)^{1/2}$ and volume $V \sim (t_f - t)^{3/2}$, but with different prefactors, which depend on the ellipsoid aspect ratio (controlled by $\theta$, see Supplementary Figure 5). The late time theoretical prediction is compared with experiments in Figs. 3, Fig. 5ab and Supplementary Figure 7. The model agrees very well with experiments at late time ($t > t_f/2$) and captures the complete shape evolution. For $t < t_f/2$, the agreement is less

satisfactory, since sharp regions are still present and influence the evaporation. The full evaporation dynamics can however be easily obtained with finite element simulations (see Methods). We extract half of the initial pointy ice drop shape by image processing and use it as an initial condition for the simulation. Figure 5a,b shows the results of the simulation for the drop volume, radius and height. The agreement is excellent and even the drop profiles at various times (Fig. 5c) are well reproduced. Interestingly, if we now look at the influence of the drop contact angle, we observe a non-monotonic dependence of the evaporation rate on the drop aspect ratio, which is captured by our model (see Supplementary Figure 7). For a given volume, temperature and humidity, hemispherical ice drops are the slowest to evaporate. Therefore, we conclude that the droplet curvature also plays a role at the scale of the drop itself, since elongated drops with points of high curvature evaporate faster.

**Simulations of evaporating snowflakes**. The study of pointy ice drops provides us the ingredients to understand the evaporation of snowflakes. However, their geometry is more complex with several sharp regions next to each other. It was shown for arrays of evaporating drops that to first order, they do not influence each other if they are more than their characteristic size apart[29]. This is not the case for the branches of the snowflakes, which are very close and interact with each other. Thus, the problem cannot be solved analytically, but we can nonetheless solve Eqs. (1) and (2) numerically using finite element simulations, which include these interactions (see Methods). We extract the initial projected snowflake shape by image processing and assume a uniform thickness as a first approximation. Then assuming typical experimental parameters, we compute the snowflake shapes at later times (Fig. 1b and Supplementary Movie 1). Because the shape evolution is governed by the geometry, this is sufficient to perfectly reproduce the snowflake shapes at various times, confirming that the very specific time evolution comes solely from the singular nature of the diffusion equation close to sharp points. Therefore, even more complex objects such as snow aggregates could be simulated, provided the full three-dimensional (3D) initial shape is known.

**Application to dissolving solids**. Our findings are relevant for studies of the ageing of snow or de-icing, but the methodology presented here is general to all free-boundary diffusive processes and can also be applied to the dissolution of solids, for instance to control the smoothness of surfaces during synthesis or to optimise crystalline shapes for drug dissolution (see Supplementary Note 2 and Supplementary Figure 8).

## Methods
**Experimental design**. The relative humidity (RH) is set between 1 and 7% by gently blowing dry nitrogen into an acrylic box housing the experiment ($0.4 \times 0.5 \times 0.7$ m) and monitored with a thermo-hygrometer Testo 645 (accuracy $\pm$ 2%). Pointy ice drops are produced by depositing small volumes ($V \sim 5\ \mu L$) of de-ionised water on clean surfaces of various wettability (and roughness) held at temperature $T_s$ by a water-cooled Peltier system (Anton Paar TEK 150P-C). The surfaces include the (rough) Peltier surface (contact angle $\theta \sim 70°$), thermal grease ($\theta \sim 95°$) and treated microscope coverslips (Menzel–Gläser #1) glued with thermal grease. Superhydrophobic surfaces ($\theta \sim 120°$) are obtained by a soot-layer deposition, followed by a silica coating[30], hydrophobic ones ($\theta \sim 85°$) by a Parylene-C coating (SCS Labcoter PDS 2010); hydrophilic ones are glass ($\theta \sim 50°$) or plasma-treated glass ($\theta \sim 20°$, Diener Zepto). To have a better control over our initial ice drop shape and volume, we avoid long supercooling periods where the drop evaporates by applying a cold shock. We ramp the surface temperature to its minimal value $T_s = -25\ °C$ to quickly freeze the drop, then as soon as the drop has started to freeze, the temperature is set to the desired value between $-15$ and $-3\ °C$.

The time evolution of the drop profile is recorded with a Canon EOS 600D camera mounted with a high-magnification objective (Navitar). The images are then analysed using ImageJ and Matlab. The tip profile is extracted with a custom sub-pixel edge detection code and fitted with an hyperbola to measure the

curvature (typical accuracy on the radius of curvature $\pm$ 3–10 μm). The full-drop profile is extracted with regular edge detection, the drop volume is measured by numerically integrating the profile (assuming axisymmetry) and the contact angle with the tangent method.

We convert our experimental parameters $T_s$, RH and room temperature $T_0 \approx$ 23–24 °C into the one used in Eqs. (1) and (2) with the following procedure. The vapour concentration at the ice surface $\rho_{sat}(T_s)$ and the one far from the drop $\rho_\infty =$ RH$\rho_{sat}(T_0)$ are obtained from the saturation pressure with the ideal gas law, the later being calculated using Eqs. (7) and (10) of ref.[25] For the water–air diffusion coefficient, we assume a constant value $D(T_0)$ that we calculate with the equation in ref.[31], figure 78. The ice density is taken as $\rho_{ice} = 918.9$ kg m$^{-3}$.

**Model assumptions**. When deriving Eqs. (1) and (2) with the bulk saturation pressure $\rho_{sat}(T_s)$, we have used the usual assumptions from the liquid droplet evaporation literature, which should also be valid here as the phenomenon is very similar. We have assumed a quasi-static process, neglected convection as it only applies in the presence of wind or very large drops[29], kinetic effects and the Kelvin effect, as it only applies to microscopic drops[32]. For ice surfaces, the presence of a microscopic amorphous layer at the surface[33–35] could also lead to a slight change in the saturation pressure. However, as this pre-melted layer is also present in previous studies measuring $\rho_{sat}$, this effect should already be implicitly taken into account: if temperature induced thickness variations in the layer, change the saturation pressure here; it should do so in all previous literature, and so is automatically included in the thermodynamic variation of the saturated vapour pressure with temperature (Eq. (7) of Ref.[25]). Finally, we assumed for simplicity a constant diffusion coefficient and that the ice surface was at the substrate temperature, thus neglecting thermal gradients (including the effect of evaporative cooling). This assumption being usually reasonable for liquid drops, except on heated substrates[36] or for very volatile liquids[37], we expect it to be even more valid for ice drops, as the ice thermal conductivity is about 4 times higher than water and 10 times higher than most organic liquids. The heat from the substrate is thus transported more efficiently to the ice surface. Moreover, the latent heat loss being proportional to the evaporation rate, since our cold ice evaporates at a much slower pace than room temperature water, we expect a much smaller evaporative cooling. At the tip though, the evaporation rate diverge for an infinitely sharp tip and thus the evaporative cooling with it. However, this occurs for a very short time only as the singularity quickly regularises itself, probably explaining why we do not need to take it into account to describe our data within our spatio-temporal resolution. We further checked the validity of the constant temperature assumption with a thermal camera (Flir C3) and found no gradients within the camera accuracy ($\pm$2 °C and $\pm$100 μm).

**Simulations**. Two-dimensional (2D) axisymmetric and 3D finite element simulations are performed with the commercial software COMSOL 5.2a. We assume that evaporation is quasi-static, isothermal and purely diffusive. We thus solve the coupled Eqs. (1) and (2) for the vapour concentration and interface movement. The mesh deformation is done with the arbitrary Lagrangian–Eulerian formulation (moving mesh ALE module with Yeoh smoothing). As the mesh quality degrades during the process, we included a small stabilising term, which penalises the formation of local curvature artefacts at the boundaries (moving boundary smoothing option), and checked that it did not influence the outcome (see Supplementary Figure 2A). Nonetheless, for complex shapes such as the snowflakes the domain needed to be re-meshed manually during the simulation.

## Data availability
The data that support the findings of this study are available from the corresponding author upon reasonable request (e.a.m.jambonpuillet@uva.nl).

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

## Acknowledgements

We thank L. Dupin for her assistance with preliminary experiments and K. Libbrecht for providing the snowflakes pictures.

## Author contributions

E.J.-P. performed the research, E.J.-P., N.S. and D.B. designed the research and wrote the paper.

## Additional information

**Competing interests:** The authors declare no competing interests.

