## [Peer Review File · Nature Communications]

Reviewers' comments:

Reviewer #1 (Remarks to the Author):

Report on *Singular crystals: sublimation of ice and snow*
by Jambon-Puillet *et al.*

Under consideration for Nature Communications

This paper presents a study of sublimation of snowflakes and of pointy ice drops, under dry and relatively cold conditions. The study aims to demonstrate the importance of sublimation in the rounding (tip smoothing) of sharp edges in ice or snow, which is a general phenomenon in diffusive processes especially in heat and mass transfer. Besides realistic numerical simulations of a sublimating snowflake, the authors chose a sessile ice drop as a simpler model experimental system, since water drops develop a tip at their summit after freezing, due to the (unusual) expansion of water upon freezing. Typical experiments allowed to extract the curvature of the tip versus time, over two decades, with great accuracy. The results are in perfect agreement with a purely diffusive theory. The experiments were repeated for several drop initial contact angle and within a range of temperature from -15°C to -3°C . Also, in later stages, the ice drop recedes on the surface at almost constant contact-angle.

Overall, the paper is clear and well-written. The methods are rigorously detailed in the S.I. and the images are of good quality.

Therefore, although the results do not show very surprising behavior or a very original phenomenon, the present study shows the important role of sublimation in the smoothing of sharp shapes in ice and snow. This is potentially of great interest in geophysical situations where spiky structures form under dry atmosphere and low enough temperature. There are a couple of points the authors should add or clarify in order to make the paper publishable in *Nature Communications* :

- I wonder how the role of the pre-melted layer of amorphous ice (evidenced for instance in Golecki & Jaccard, J. Phys. C 11, 4229 (1978); Engelmann *et al.*, Phys. Rev. Lett. 92, 205701 (2004) or Doppenschmidt and Butt, Langmuir 16, 6709 (2000)) can influence here the sublimation rate. Could the influence of T on the thickness of this pre-melted layer explain the evolution of the coefficient β with T (figure S8-B) ? Please comment.
- Figure 2 and Figure S4 show a quasi-constant evolution of the ice drop contact angle. The tip smoothing could also happen near the contact-line and hence, the value of the angle should be around 90° , which seems plausible according to the measurements. Could the authors show the plot of the final contact-angle versus initial contact angle and comment on this ?
- In the formation of ice penitentes, the recent study by Claudin *et al.* (Phys. Rev. E 92, 033015 (2015)) should be cited, as it emphasized the role of sublimation in the formation of undulatory and spiky structures in the environment.

Reviewer #2 (Remarks to the Author):

The authors discuss evaporation of snow-flakes and ice drops. They compared the evaporation dynamics resulting from finite element calculations with their experimental data. In particular, they investigated the profile of ice tips and showed that the profiles are self-similar. This holds independent of the hydrophobicity of the surface. Furthermore, the authors numerically calculated (Comsol package) the evaporation profile of a snowflake. They conclude that the evaporation of snow-flakes is governed by the domains with the highest curvature. Thus, the shape evolution of snow-flakes is determined by its geometry.

Snow-flakes are common model systems of fractal growth. This implies that the shape needs to be self-similar. From that point, I wasn't surprised at all to read that the authors found self-similarity in their geometries. It's also clear that sharp edges foster evaporation.

However, to my surprise, as of the time of writing, most researchers appear to have focused on investigating the growth dynamics of snow-flakes or ice droplets but not on their melting or sublimation dynamics. Indeed, I didn't find anything on the analysis of the shape of sublimating snow-flakes. Although the authors observed what I would have expected, it is important to have it proven. Therefore I do recommend publication of the manuscript after a few modifications.

The authors need to remove the terms "surprisingly,...", as these observed phenomena should not be unexpected. For example: The authors wrote that "the fact that the ice contact line retracts as for a liquid is surprising since the anchoring of the drop to the rough surface is very strong". I disagree. I think it would be surprising if it would be vice versa. According to molecular dynamic simulations, the kinetic energy (thus degree of freedom) of undercooled water and ice is very similar. See for example: Ramírez et. al. The Journal of Chemical Physics, 141, 204701 (2014) and references therein.

On page 3 line 175-178 the authors wrote that "elongated drops with points of high curvature evaporate faster". This seems to be one of the authors' main conclusions. Wasn't that clear from the very beginning?

Please explain why the assumption of a uniform thickness is considered reasonable.

I expect that the evaporation of the different "branches" of the snowflake influence each other resulting in reduced evaporation velocity as a snowflake also has sharp tips within its profile (see Fig. 1 or <https://www.chemistryworld.com/news/what-makes-a-snowflake-special/3008386.article>)

Please argue why it's safe to ignore evaporation induced cooling of the ice tips? See for example: PHYSICAL REVIEW E 89, 042402 (2014) for a discussion of the temperature profile of evaporating water drops. Can you comment on how the temperature profile close to the sharp tip varies during evaporation?

Supporting information:

I really would appreciate more detailed derivations. After reading the first few pages I gave up. Add a , b and α , η , and ϕ to Fig. S1 or add another sketch.

Eq. S1: Shouldn't it be $\frac{\Delta\rho}{\Delta r}$? Please explain. Add how ρ depends on η .

Are the n in $\frac{\Delta\rho}{\Delta n}$ and the parameter n on page S2 identical?

Page S2: β is only defined several lines later. Change the order to improve readability.

S2 -> S3: Add more details. What are the integration constants?

We would like to thank the referees for their careful reading of the manuscript and for their constructive comments which substantially helped improving the quality of the paper. We have taken into account all the referees' comments in the revised version of the manuscript and supporting information. We provide below a detailed answer (in green) to each point raised by the referee as well as a version of the revised manuscript where subsequent changes in the text in are highlighted in red. We also made minor changes to comply with Nature Communications' format but did not highlight them as they do not impact the scientific content. The numbering of references, figures and equations here corresponds to the numbering of the first submission.

Referee 1:

This paper presents a study of sublimation of snowflakes and of pointy ice drops, under dry and relatively cold conditions. The study aims to demonstrate the importance of sublimation in the rounding (tip smoothing) of sharp edges in ice or snow, which is a general phenomenon in diffusive processes especially in heat and mass transfer. Besides realistic numerical simulations of a sublimating snowflake, the authors chose a sessile ice drop as a simpler model experimental system, since water drops develop a tip at their summit after freezing, due to the (unusual) expansion of water upon freezing. Typical experiments allowed to extract the curvature of the tip versus time, over two decades, with great accuracy. The results are in perfect agreement with a purely diffusive theory. The experiments were repeated for several drop initial contact angle and within a range of temperature from -15°C to -3°C . Also, in later stages, the ice drop recedes on the surface at almost constant contact-angle.

Overall, the paper is clear and well-written. The methods are rigorously detailed in the S.I. and the images are of good quality.

Therefore, although the results do not show very surprising behavior or a very original phenomenon, the present study shows the important role of sublimation in the smoothing of sharp shapes in ice and snow. This is potentially of great interest in geophysical situations where spiky structures form under dry atmosphere and low enough temperature.

We thank the referee for the careful reading of our manuscript, and the very helpful comments. We would like to point out that there are several results that are surprising or original and that we believe will be of interest to physicists, hydrodynamicists and mathematicians. That the tip of the pointy ice drop is a singularity in the Laplace equation is indeed well known. However, here unlike almost all of the literature on the subject the interface is free to move and so a more complex free boundary problem emerges, giving rise to our new and original mathematical treatment. That this works so well is actually surprising, because it shows that the underlying crystalline structure is irrelevant. Indeed, the growth is dominated by the crystalline structure that results in the branched structure of the snowflakes. For the sublimation the crystal structure is not important, contrary to what is argued in the single paper on the subject in the literature (Nelson, J. Sublimation of Ice Crystals, J. Atmospheric Sci 55, 910-919 (1998)). We have underlined all of this in the revised version of the manuscript.

There are a couple of points the authors should add or clarify in order to make the paper publishable in Nature Communications :

- I wonder how the role of the pre-melted layer of amorphous ice (evidenced for instance in Golecki & Jaccard, J. Phys. C 11, 4229 (1978); Engelmann et al., Phys. Rev. Lett. 92, 205701 (2004) or Doppenschmidt and Butt, Langmuir 16, 6709 (2000)) can influence here the sublimation rate. Could the influence of T on the thickness of this pre-melted layer explain the evolution of the coefficient β with T (figure S8-B) ? Please comment.

As mentioned in the SI, there are many phenomena we neglected that could explain the slightly higher experimental β compared with the purely diffusive theory at constant temperature. Thermodynamically, the presence of a different surface layer could lead to a slight change in the saturation pressure. We think that this is in fact already included in the tabulated values we used as the data from the previous literature implicitly takes such a layer into account: if thickness variations in the layer changes the saturation pressure here, it should do so in all previous experiment, and so is automatically included in the thermodynamic variation of the saturated vapor pressure with temperature. We have added a paragraph in the first section of the SI to detail the model's assumptions and included the amorphous ice layer to this discussion (including these references).

- Figure 2 and Figure S4 show a quasi-constant evolution of the ice drop contact angle. The tip smoothing could also happen near the contact-line and hence, the value of the angle should be around 90° , which seems plausible according to the measurements. Could the authors show the plot of the final contact-angle versus initial contact angle and comment on this ?

Indeed, as was already mentioned in the paper, a similar smoothing happens also near the drop edge. This is the reason why during the sublimation all our drops end up with a contact angle of 90° (within the experimental uncertainties) which allows us to model them as half ellipsoids in the SI. Nonetheless, quantitatively there might be some differences since the geometry is mostly 2D here. We expect that a very similar derivation considering hyperbolas instead of hyperboloids can capture this dynamics.

We have underlined this once more in the main text of the revised manuscript and added the plot of the final angle versus initial angle in Fig. S4.

- In the formation of ice penitentes, the recent study by Claudin et al. (Phys. Rev. E 92, 033015 (2015)) should be cited, as it emphasized the role of sublimation in the formation of undulatory and spiky structures in the environment.

We thank the referee for pointing this out to us and have added the reference. To show how non-trivial the tip smoothing is, we added a sentence describing seemingly similar cases where more pointy structures forms instead of less pointy ones (the penitentes, solids dissolving under flow: Ristroph, L., et al PNAS 109, 19606-19609 (2012) and Nakouzi, E., Goldstein, R. E. & Steinbock, O. Langmuir 31, 4145-4150 (2015)).

Referee 2:

The authors discuss evaporation of snow-flakes and ice drops. They compared the evaporation dynamics resulting from finite element calculations with their experimental data. In particular, they investigated the profile of ice tips and showed that the profiles are self-similar. This holds independent of the hydrophobicity of the surface. Furthermore, the authors numerically calculated (Comsol package) the evaporation profile of a snowflake. They conclude that the evaporation of snow-flakes is governed by the domains with the highest curvature. Thus, the shape evolution of snow-flakes is determined by its geometry.

Snow-flakes are common model systems of fractal growth. This implies that the shape needs to be self-similar. From that point, I wasn't surprised at all to read that the authors found self-similarity in their geometries. It's also clear that sharp edges foster evaporation.

However, to my surprise, as of the time of writing, most researchers appear to have focused on investigating the growth dynamics of snow-flakes or ice droplets but not on their melting or sublimation dynamics. Indeed, I didn't find anything on the analysis of the shape of sublimating

snow-flakes. Although the authors observed what I would have expected, it is important to have it proven. Therefore I do recommend publication of the manuscript after a few modifications.

As already mentioned to referee 1, there are several results that are surprising or original and that we believe will be of interest to physicists, hydrodynamicists and mathematicians.

- 1) That the tip of the pointy ice drop is a singularity in the Laplace equation is indeed well known. However, here unlike almost all of the literature on the subject the interface is free to move and so a more complex free boundary problem emerges, giving rise to our new and original mathematical treatment.
- 2) That this works so well is actually surprising, because it shows that the underlying crystalline structure is irrelevant. Indeed, the growth is dominated by the crystalline structure that results in the branched structure of the snowflakes. For the sublimation the crystal structure is not important, contrary to what is argued in the single paper on the subject in the literature (Nelson, J. Sublimation of Ice Crystals, J. Atmospheric Sci 55, 910-919 (1998)). We have underlined all of this in the revised version of the manuscript. Thus what we find shows that crystallization and sublimation are in fact very different.
- 3) The observation that snowflakes are fractal is in fact different from our observation that a single tip sublimates in a self-similar fashion. The fractality only tells something about the structure, whereas the similarity solution that we find for the singular pointy regions gives the dynamics: it allows us to predict how the curvature changes in time.

We have underlined all three points in the revised version of the manuscript, to make this very clear.

The authors need to remove the terms “surprisingly,...”, as these observed phenomena should not be unexpected. For example: The authors wrote that “the fact that the ice contact line retracts as for a liquid is surprising since the anchoring of the drop to the rough surface is very strong”. I disagree. I think it would be surprising if it would be vice versa. According to molecular dynamic simulations, the kinetic energy (thus degree of freedom) of undercooled water and ice is very similar. See for example: Ramírez et. al. The Journal of Chemical Physics, 141, 204701 (2014) and references therein.

We have removed the multiple mentions of “surprisingly” from the text.

For the contact line, since for liquid droplets which are easy to remove from the interface the contact line stays pinned, it does not seem intuitive to us that the ice droplet contact line appears mobile over long times as it is very strongly anchored onto the surface (if one wants to remove it, it takes quite some force). We found difficult to enter in all the discussions about the differences/peculiarities of ice and supercooled water in a short paper about the sublimation of ice, but now briefly discuss Ramirez et al.'s results when we discuss the similarity in evaporation rate between water and ice.

On page 3 line 175-178 the authors wrote that “elongated drops with points of high curvature evaporate faster”. This seems to be one of the authors’ main conclusions. Wasn’t that clear from the very beginning?

Although this is what one would intuitively expect, here we provide the first detailed measurements of this effects and provide a simple analytical model to quantify this effect (see SI equation S.12 and Fig. S9) which agrees with both experiments and simulations.

Please explain why the assumption of a uniform thickness is considered reasonable.

Since we do not have information about the 3D shape of the snowflake in Fig. 1, assuming uniform thickness was the simplest and thus first thing we tried. The excellent agreement shows that this

simplest assumption suffices here. For more complicated snowflakes or aggregates the full 3D shape would indeed be required. We have added in the text that this is the simplest case one can consider and that for more complex objects 3D effects need to be taken into account.

I expect that the evaporation of the different “branches” of the snowflake influence each other resulting in reduced evaporation velocity as a snowflake also has sharp tips within its profile (see Fig. 1 or <https://www.chemistryworld.com/news/what-makes-a-snowflake-special/3008386.article>)

They do influence each other and this is explicitly taken into account in the finite element simulations. This has been investigated in the context of collections of evaporating drops (Carrier et al JFM 2016) and the conclusion is that to first order if the structures are more than their characteristic size apart, the influence on the evaporation is small. As this is not the case for the snowflakes, the effect is probably important here. We discuss this now in the revised version, have added the reference and underline once more the excellent agreement with the numerical simulation.

Please argue why it’s safe to ignore evaporation induced cooling of the ice tips? See for example: PHYSICAL REVIEW E 89, 042402 (2014) for a discussion of the temperature profile of evaporating water drops. Can you comment on how the temperature profile close to the sharp tip varies during evaporation?

Again, we started with the simplest hypotheses: purely diffusive evaporation at constant temperature and it turned out to be sufficient to describe our experiments quite well. Moreover, we tried to look for temperature gradients with an infrared thermal camera (Flir C3). Within the accuracy of the camera ($\pm 2^\circ\text{C}$ and $\pm 100\ \mu\text{m}$) we did not observe any. This has been added to the revised version of the SI.

There are a few qualitative reasons that explains why we expect it to be less important for ice than for liquid droplets. First, the ice thermal conductivity is about 4 times higher than water and 10 times higher than most organic liquids, thus the heat from the substrate is transported more efficiently to the ice surface. Then as the latent heat loss is proportional to the evaporation rate, temperature effects are predominant for liquid drops on heated substrates (like in your ref) or very volatile liquids (E. Jambon-Puillet et al. *J. Fluid Mech.*, 844, 817-830 (2018)). Our cold ice evaporating at a much slower rate, we thus expect a much smaller evaporative cooling.

Of course at the tip though, the evaporation rate diverge for an infinitely sharp tip and thus the evaporative cooling with it. However, this occurs for a very short time only as the singularity regularizes itself, probably explaining why we don’t need to model it to describe our data within our spatio-temporal resolution.

All this is discussed in the revised version of the SI.

Supporting information:

I really would appreciate more detailed derivations. After reading the first few pages I gave up. Add a , b and α , η , and ϕ to Fig. S1 or add another sketch.

We did our utmost to make the derivations more accessible. We have added a new sketch in Fig. S1 to define a , b and α . As ξ , η and ϕ are standard notations for the prolate spheroidal coordinates (also used in SI ref [3]), we provide a link to the Wolfram Mathworld page where all the notations are explained with schematics. We further added the reference (J. Lekner, *European Journal of Physics* 27, 87 (2006)) which essentially reproduces the calculations of SI ref [3,4] in a very pedagogical way. Referee 1 also thought that “The methods are rigorously detailed in the S.I.”, but we have tried to do even better.

Eq. S1: Shouldn't it be $\frac{\delta\rho}{\delta r}$? Please explain. Add how ρ depends on η . Are the n in $\frac{\delta\rho}{\delta n}$ and the parameter n on page S2 identical?

It is indeed $\frac{\delta\rho}{\delta n}$ where n here denotes the normal vector to the hyperboloidal surface ξ_s so it is the gradient of the field in the normal direction (or normal derivative) as in Eq. (2) but with a slightly different notation (see also SI refs [3]). We now mention it explicitly and have adjusted Eq. (2) to point out that the normal derivative must be evaluated on the solid surface.

For an hyperboloid ρ does not depend on η (or ϕ)! This means that equipotential are also hyperboloidal surfaces (see SI ref [3][4]).

We thank the referee for highlighting this bad choice of variable names, the two parameters are not identical, the first one being the normal vector while the second one is simply a fitting parameter to adjust for the crude asymptotic matching. We changed the latter to "q" to avoid confusion. We further changed the name of the integration constants at the very beginning to C_1 and C_2 avoiding the confusion with the prefactor A .

Page S2: β is only defined several lines later. Change the order to improve readability.

We thank the referee for pointing this out and now define β just after its first occurrence in the text.

S2 -> S3: Add more details. What are the integration constants?

We have added two intermediate steps between S2 and S3 and hope that the calculation is now very clear. As the procedure is strictly identical between S6 and S7 with only a more cumbersome integral to calculate we did not repeat these steps here.

REVIEWERS' COMMENTS:

Reviewer #1 (Remarks to the Author):

I read the author's reply to my comments and suggestions, and the new version of the main manuscript and SI, and I think they addressed thoroughly my comments and those of the other referee. Therefore, I think the paper is publishable in its current state.

Reviewer #2 (Remarks to the Author):

The authors discussed in more detail the assumptions and limitations of their work. These are nicely summarized in the first paragraph of the SI. I like that the authors stressed even more that sublimation has nothing to do with the underlying crystal structure, the analogy between the Laplace equation and electrostatics and the similarities between ice and super cooled water molecules.

The more detailed discussion of the assumptions and limitations helps to identify the novel and surprising aspects of the manuscript and identifies interesting research directions. The authors also improved the readability of the SI for non-experts.

Minor: The authors wrote: "Our new similarity solution...": please delete „new“. Results published in Nature communications should be "new" by definition.

I'm happy with the revised version of the manuscript and recommend publication.

Referee 1:

I read the author's reply to my comments and suggestions, and the new version of the main manuscript and SI, and I think they addressed thoroughly my comments and those of the other referee. Therefore, I think the paper is publishable in its current state.

We would like to thank the referee for his careful reading of the revised manuscript and for its positive comments.

Referee 2:

The authors discussed in more detail the assumptions and limitations of their work. These are nicely summarized in the first paragraph of the SI. I like that the authors stressed even more that sublimation has nothing to do with the underlying crystal structure, the analogy between the Laplace equation and electrostatics and the similarities between ice and super cooled water molecules.

The more detailed discussion of the assumptions and limitations helps to identify the novel and surprising aspects of the manuscript and identifies interesting research directions. The authors also improved the readability of the SI for non-experts.

We thank the referee for carefully reading our revised manuscript and for these positive comments. We have now included the assumption and limitation paragraph from the SI directly in the methods of the main text to improve the readability of the paper.

Minor: The authors wrote: "Our new similarity solution...": please delete „new“. Results published in Nature communications should be "new" by definition.

We have removed the word "new" in this sentence.

I'm happy with the revised version of the manuscript and recommend publication.